# Evaluation of Spleen Stiffness in Young Healthy Volunteers Using Magnetic Resonance Elastography

**DOI:** 10.3390/diagnostics13172738

**Published:** 2023-08-23

**Authors:** Marzanna Obrzut, Vitaliy Atamaniuk, Richard L. Ehman, Meng Yin, Marian Cholewa, Krzysztof Gutkowski, Wojciech Domka, Dorota Ozga, Bogdan Obrzut

**Affiliations:** 1Institute of Health Sciences, Medical College, University of Rzeszow, Warzywna 1a, 35-310 Rzeszow, Poland; marzannaobrzut@gmail.com (M.O.);; 2Department of Biophysics, Institute of Physics, College of Natural Sciences, University of Rzeszow, Prof. Stanisława Pigonia Str. 1, 35-310 Rzeszow, Poland; vitaliya@dokt.ur.edu.pl (V.A.); mcholewa@ur.edu.pl (M.C.); 3Department of Radiology, Mayo Clinic, 200 First St. SW, Rochester, MN 55905, USA; 4Institute of Medical Sciences, Medical College, University of Rzeszow, Rejtana 16C, 35-959 Rzeszow, Poland; kgutski@intertele.pl; 5Department of Otolaryngology, Institute of Medical Sciences, Medical College, University of Rzeszow, Rejtana 16C, 35-959 Rzeszow, Poland; w.domka@gazeta.pl; 6Department of Obstetrics and Gynecology, Institute of Medical Sciences, Medical College, University of Rzeszow, Rejtana 16C, 35-959 Rzeszow, Poland

**Keywords:** magnetic resonance elastography, tissue stiffness, normative values

## Abstract

Purpose: Magnetic resonance elastography (MRE) has been established as the most accurate noninvasive technique for diagnosing liver fibrosis. Recent publications have suggested that the measurement of splenic stiffness is useful in setting where portal hypertension may be present. The goal of the current study was to compile normative data for MRE-assessed stiffness measurements of the spleen in young adults. Materials and Methods: A total of 100 healthy young Caucasian volunteers (65 females and 35 males) in the age range of 20 to 32 years were enrolled in this study. The participants reported no history of chronic spleen and liver disease, normal alcohol consumption, and a normal diet. The MRE data were acquired by using a 1.5 T whole-body scanner and a 2D GRE pulse sequence with 60 Hz excitation. Spleen stiffness was calculated as a weighted mean of stiffness values in the regions of interest manually drawn by the radiologist on three to five spleen slices. Results: Mean spleen stiffness was 5.09 ± 0.65 kPa for the whole group. Male volunteers had slightly higher splenic stiffness compared to females: 5.28 ± 0.78 vs. 4.98 ± 0.51 kPa, however, this difference was not statistically significant (*p* = 0.12). Spleen stiffness did not correlate with spleen fat content and liver stiffness but a statistically significant correlation with spleen volume was found. Conclusions: The findings of this study provide normative values for 2D MRE-based measurement of spleen stiffness in young adults, a basis for assessing the value of this biomarker in young patients with portal system pathologies.

## 1. Introduction

Magnetic resonance elastography (MRE) is an innovative diagnostic technique used to assess the stiffness of soft tissues in a non-invasive manner. MRE was developed by researchers at the Mayo Clinic in Rochester, USA, during the 1990s [1,2]. Initial scientific papers presenting the theoretical foundations of MRE, along with results from examinations using phantom and tissue samples, were published in 1995 in Science and in 1996 in Nature Medicine [1,2]. The first human studies on measuring liver stiffness using MRE were presented at the Annual Meetings of the International Society for Magnetic Resonance in Medicine (ISMRM) in 2004 and 2005 [3,4]. Subsequent research publications have further confirmed the effectiveness of MRE in diagnosing liver fibrosis [5,6,7,8,9].

From a physical perspective, MRE involves three steps. The first step is to induce shear waves in the targeted area. This is achieved by applying vibrations generated by an active acoustic driver located outside the diagnostic room. Typically, the vibration frequency ranges from 20 to 100 Hz [10]. The acoustic waves produced are transmitted to a disc-shaped passive acoustic driver in contact with the tissue via a connecting polyvinyl chloride tubing. 

The second step is to image the shear waves propagating through the human body using MRI. In MRE, tissue movement induced by the driver is measured using a specific phase-contrast MRI sequence, such as gradient recalled echo or spin-echo echo-planar imaging. Motion-encoding gradient pairs (MEGs) are used to encode tissue displacements [10]. The MEGs can be applied in any direction. 

MRE typically involves multiple image acquisitions at different time points of the vibration cycle to obtain 3 to 8 phase offsets between MEGs and mechanical oscillation equally distributed over the course of the motion cycle [10]. This allows for the visualization of shear waves propagating through the body and enables MRE to encode tissue displacements at the nanometer or micrometer level.

The final component of MRE is the application of an appropriate inversion algorithm to the images depicting shear wave propagation in the tissues. In this step, stiffness maps, also known as elastograms, are generated. Stiffness is usually reported as a shear modulus magnitude (|G*| or µ) or shear wave speed (SWS) [11].

The MRI scanner equipped with a commercially available, FDA-approved MRE setup automatically applies a multimodel direct inversion (MMDI) algorithm to the wave propagation images and generates elastograms. The scanner software also allows for the overlaying of the stiffness maps onto the corresponding magnitude anatomical images to facilitate the exam reading process.

Another crucial step in evaluating tissue mechanical properties using MRE is selecting a region of interest (ROI) on the stiffness map. The ROI is used to calculate the average stiffness of the assessed tissue. The ROI drawn by a radiologist can have various shapes, such as oval, round, or irregular. It is also recommended to avoid organ edges, focal lesions, vessels with a diameter exceeding 3 mm, and areas with wave interference and pulsation-related artifacts when drawing ROIs to ensure reliable inversion [12]. The final stiffness measurement result is usually reported as the mean value of stiffness in the region of interest, expressed in kilopascals (kPa) or meters per second (m/s).

Currently, magnetic resonance elastography is considered the most accurate and reliable non-invasive method for diagnosing and staging liver fibrosis in patients with chronic liver disease. It has the potential to serve as an alternative to expensive and invasive liver biopsy. Moreover, MRE enables the diagnosis of liver fibrosis even at early stages, when it is still reversible [13,14,15,16,17,18,19].

With ongoing research and advancements in technology, MRE holds promise for expanding its applications to a wide range of organs and conditions, ultimately improving patient care and outcomes. Researchers are investigating the clinical utility of MRE in evaluating other organs and tissues, such as the pancreas, kidney, prostate, uterus, brain, breast, and muscles [20,21,22,23,24,25,26,27,28,29].

MRE may also be of significant help in tumor diagnostics, as the mechanical properties of healthy tissue and different types of tumors usually vary. Its ability to provide quantitative measurements of tissue properties opens up new possibilities for diagnosing and monitoring various pathologies and diseases. These potential applications of MRE are supported by evidence demonstrating excellent reproducibility and repeatability [30,31,32,33,34].

An active area of research focuses on determining the usefulness of measuring spleen stiffness using MRE for detecting portal hypertension [35,36], staging liver cirrhosis [19,37], or predicting esophageal varices in chronic liver disease [38,39,40]. According to the study of Ronot et al., spleen stiffness may be the best biomarker for identifying patients with severe portal hypertension or high-risk esophageal varices [35]. Other research revealed that the combination of spleen stiffness and liver stiffness can increase sensitivity in the prediction of advanced liver fibrosis [37]. However, there is a need to establish reference data for these studies since information on MRE-based splenic stiffness in healthy individuals is currently limited [33,38]. Therefore, the aim of this work is to compile normative data for MRE-based spleen stiffness in a group of healthy young subjects.

## 2. Methods

The Ethics Committee of the Medical Department of the University of Rzeszow approved the study protocol (Resolution No 8/10/2016) and each participant provided written informed consent before being enrolled in this study. The study protocol conformed to the ethical guidelines of the 1975 Declaration of Helsinki (6th revision, 2008).

According to a priori power analysis, a sample size of 99 participants was required for this study (α = 0.05, power of 80%; estimated effect size of 0.4). However, more volunteers were recruited since, based on our experience, we expected 15 to 25% of participants not to meet the inclusion criteria. Finally, 100 out of 115 volunteers met the eligibility criteria and were involved in this study. 

The following factors were taken into account when choosing the target group:Potential contraindications to MRI

Volunteers with pacemakers, drug infusion devices, metallic foreign bodies, or metallic ink tattoos were excluded from the study;

History of liver and spleen disease of the volunteer and his or her familyOnly participants with no history of liver and spleen disease were selected; Liver function tests results

ASPAT: 17–59 U/Lfor male and 14–36 U/L for female, ALAT: <50 U/L for male and <35 U/L for female, Lipase: 23–300 U/L, liver iron: 49–181 µg/dL for male and 37–170 µg/dL for female;

Alcohol consumption, diet type, and medicines taken by the individual

Each subject confirmed following a normal diet, consuming no more than 30 g of alcohol per day, and taking no medication;

Liver stiffness and fat fraction values

Only volunteers with normal liver stiffness < 2.9 kPa and liver fat fraction < 5% were enrolled in this study;

Body Mass Index (BMI)

Obese and severely obese participants were excluded from this study.

The MRE data were acquired at the University of Rzeszow’s Center for Medical and Natural Sciences Research and Innovation during the period from December 2018 until July 2019.

### 2.1. Study Protocol 

A 1.5-T whole-body unit MR OPTIMA MR360 Advance (GE Healthcare, Milwaukee, Wisconsin, United States) was used in this study. The foot-first supine position was chosen for the examination with a drum-like passive driver placed approximately at the spleen level over the subject’s left side of the abdomen. The actuator was secured with an elastic strap and the 8-channel torso phased array coil was positioned above. 

To acquire the spleen MRE data, a 2D modified gradient recalled echo-based MR Touch pulse sequence (GE Healthcare, Waukesha, WI, USA) with the following imaging parameters was applied: 60 Hz vibrations, 40 cm FOV, 33.3 ms TR, 20.5 ms TE, 224 × 64 acquisition matrix, 256 × 256 recon matrix size, 10 mm slice thickness, 30° flip angle, and 4 time offsets.

Eight to nine slices were imaged through the spleen during three to four breath holds at the end of expiration. The stiffness maps generated on the scanner were transferred offline for further analysis and measurements.

### 2.2. Image Analysis

The MRE data were analyzed simultaneously by an MRE technician with more than five years of experience and a radiologist with more than twenty years of experience. One or two irregular polygon ROI masks of the maximum possible area were manually drawn on three to five spleen slices on the stiffness maps (Figure 1). The weighted mean spleen stiffness value was then calculated for each subject. The spleen borders and large vessels were avoided when placing the ROIs to increase the inversion accuracy. The average size of the ROI for the spleen was 3 to 5 cm^2^.

### 2.3. Statistical Analysis

The mean (SD) values of the spleen stiffness were calculated based on the values from each slice. The Shapiro-Wilk test was performed to investigate the normality of the distribution of measurable variables. The correlation between spleen stiffness and other parameters, including spleen fat content and spleen volume was assessed by calculating Spearman rank correlation coefficients. A Mann-Whitney U-test was performed to determine whether the difference in spleen stiffness values in males and females was statistically significant.

Statistical significance was set at a *p* value < 0.05. The open-source JASP Statistical Software version 0.16.4 (University of Amsterdam, Amsterdam, The Netherlands; https://jasp-stats.org, accessed on 5 May 2023) was used for statistical modeling.

## 3. Results

A total of 15 subjects initially selected for the study did not meet all inclusion criteria, reducing the total cohort from 115 to 100 subjects. All 100 volunteers (65 females and 35 males) had successful MRE examinations without reporting any discomfort during the data acquisition. 

The calculated mean [SD] spleen stiffness value in the cohort was 5.09 [0.65] kPa (95% CI, 4.97–5.23), ranging from 3.89 to 7.24 kPa.

The BMI was in the range of 16.85–25.76 kg/m^2^ with a mean [SD] of 21.33 [2.16] kg/m^2^. The spleen volume ranged from 123.94 to 422.43 cm^3^ while the mean [SD] value was 231.13 [58.44] cm^3^.

The difference in splenic stiffness between female (mean [SD], 4.98 [0.51] kPa; range 3.89–6.03 kPa) and male volunteers (mean [SD], 5.28 [0.78] kPa; range 3.99–7.24 kPa) was not significant statistically (*p* = 0.12) (Figure 2). 

The mean [SD] spleen fat content value was 2.04 [0.39]%, ranging from 1.15% to 3.27%, and no correlation with spleen stiffness was found (rho= −0.13; *p* = 0.20) (Figure 3). Spleen stiffness did not correlate with spleen R2* (rho = 0.03, *p* = 0.74), BMI (rho = 0.09; *p* = 0.36) (Figure 4), and liver stiffness (rho = 0.01; *p* = 0.95) (Figure 5). However, spleen stiffness was found to be positively correlated with spleen volume, and this relationship was statistically significant (rho = 0.40, *p* < 0.001) (Figure 6). The results are summarized in the Table 1.

## 4. Discussion

The spleen stiffness obtained in the group of healthy volunteers ranged from 3.89 to 7.24 kPa with a mean value of 5.09 kPa. These results show that the normal spleen is stiffer than the normal liver, pancreas, prostate, and uterus [20,41,42,43,44,45].

The spleen stiffness value from our study was higher than in a group of 12 volunteers recruited by Talwalkar et al.: 3.6 ± 0.3 kPa [46]. Other research, involving sixteen volunteers, reports a mean splenic stiffness of 4.255 ± 0.625 kPa [47]. In another study, Yasar et al., examined 12 subjects (5 healthy volunteers and 7 patients with chronic liver disease) to assess inter-platform reproducibility of liver and spleen stiffness measurement by MRE [33]. The mean value of spleen stiffness was reported for the whole group and reached 7.54 to 7.91 kPa. There was no differentiation between healthy and ill participants, so the findings cannot be compared to the current results. Another retrospective study reported a mean splenic stiffness value of 5.2 ± 1.5 kPa in a sub-group of 115 patients not displaying symptoms of chronic liver disease [38]. This value is comparable to our own findings but was obtained based on a retrospective analysis of medical records. 

The splenic stiffness was higher in men than in women but this difference did not show statistical significance. Prior reports have been mixed in this regard [46,47].

In contrast to the results of Talwalkar et al. [46], the spleen stiffness obtained in the group under investigation did not correlate with liver stiffness. 

The estimated fat content in the spleen and spleen R2* also did not impact its stiffness in healthy volunteers. On the other hand, we observed a statistically significant correlation between spleen stiffness and spleen volume in healthy volunteers. 

Alternative shear wave-based methods for elasticity assessment of the spleen are ultrasound-based transient elastography (TE), point shear wave elastography (pSWE), and 2D and 3D shear wave elastography (SWE) [48,49]. The available data on splenic stiffness in healthy volunteers derived from US-based elastography are mixed. In a study by Arda et al., a mean SWE-based spleen stiffness value of 2.9 ± 1.8 kPa was revealed [50]. Takuma et al., obtained spleen stiffness of 2.16 (IQR, 1.99–2.26) m/s measured by acoustic radiation force impulse shear wave imaging in a group of 16 healthy volunteers [51]. Leung et al., reported a mean value of SWE-based spleen stiffness of 17.3 ± 2.6 kPa in a group of 171 healthy individuals [52]. The mean value of splenic stiffness measured by Pawluś et al., was determined as 16.6 ± 2.5 kPa [53]. Giuffrè et al., using point shear wave elastography, obtained a mean splenic stiffness value of 18.14 ± 3.08 kPa [54]. There were no statistically significant differences in stiffness values between sexes [50,52,53,54]. Also, no correlation between splenic stiffness and spleen size was found [53,54]. 

It is worth emphasizing that ultrasound-based elastography techniques differ from each other and MRE in terms of technological and physical details as well as methods of data acquisition [53]. Hence, there is no possibility to compare or extrapolate the findings obtained by using different diagnostic modalities. 

The presented study has several limitations. In this project, we focused on assessing the spleen stiffness of healthy young adults, therefore, the age range of the subjects was relatively narrow. Future studies including individuals within a wider age range should be conducted to verify the results. The study included exclusively Caucasian volunteers, which might be also perceived as a limitation. Finally, the study was conducted with a 2D MRE technique, which has known limitations compared with 3D MRE for splenic stiffness measurement but the latter technique is not yet widely available.

The current research is the largest prospective study compared with all other reports on this topic. Another strength of the presented study is the meticulous selection of participants, which increases the credibility of the obtained findings.

The findings of this study provide normative values for 2D MRE-based measurement of spleen stiffness, a basis for assessing the value of this biomarker in young patients with portal system pathologies.

## Figures and Tables

**Figure 1 diagnostics-13-02738-f001:**
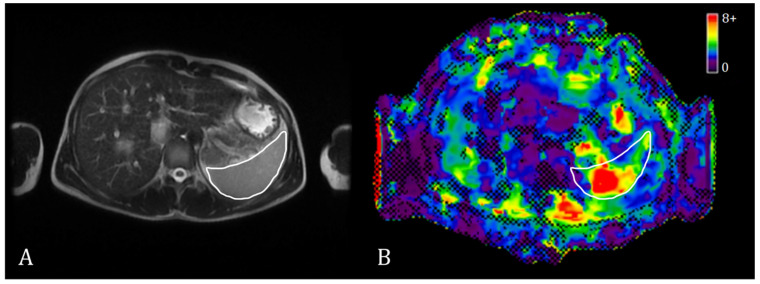
Exemplary T2w MR image (**A**) and a stiffness map with a confidence mask (**B**) in spleen MRE. The spleen is outlined with a white contour.

**Figure 2 diagnostics-13-02738-f002:**
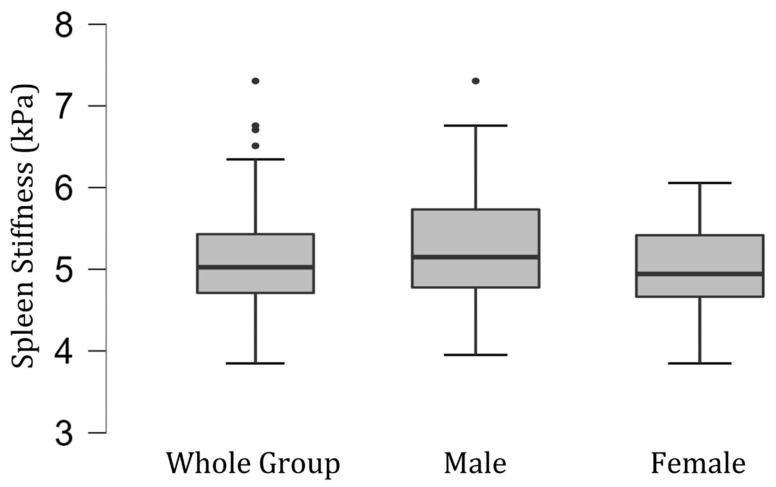
Box and whisker plots of the distribution of measured spleen stiffness values (in kilopascals) in the group of 100 healthy volunteers—35 men and 65 women.

**Figure 3 diagnostics-13-02738-f003:**
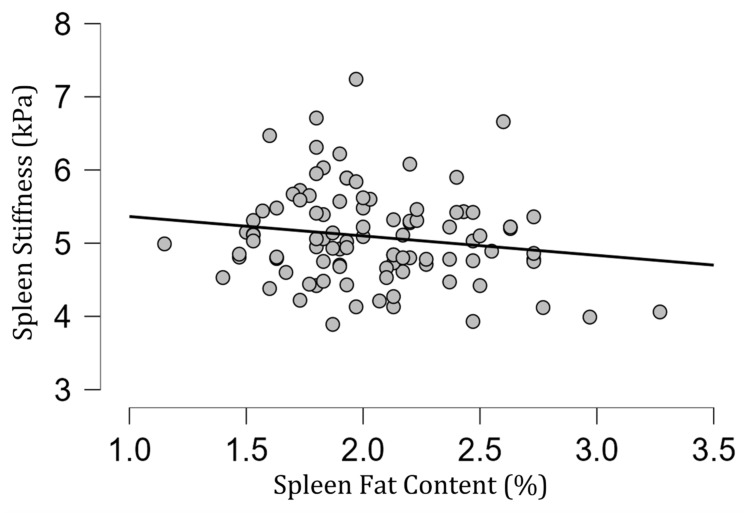
A scatter plot showing the correlation between spleen stiffness (in kilopascals) and spleen fat content.

**Figure 4 diagnostics-13-02738-f004:**
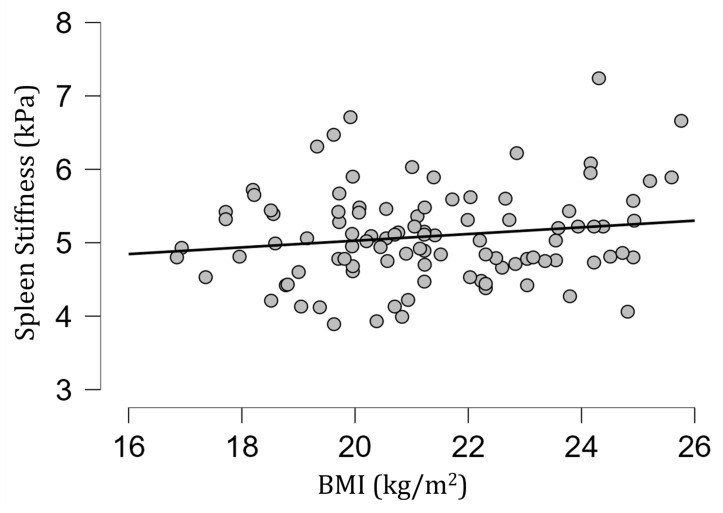
A scatter plot showing the correlation between spleen stiffness (in kilopascals) and BMI. No significant correlation was observed.

**Figure 5 diagnostics-13-02738-f005:**
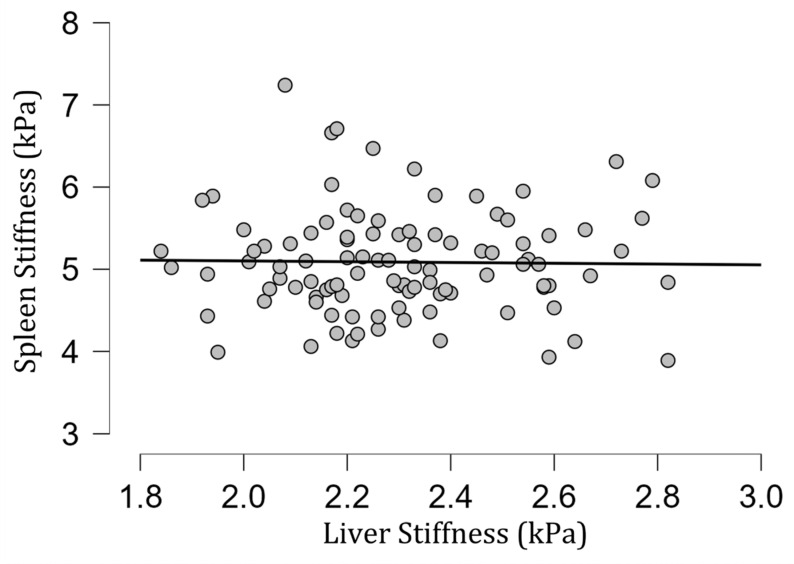
A scatter plot showing the correlation between spleen and liver stiffness (in kilopascals).

**Figure 6 diagnostics-13-02738-f006:**
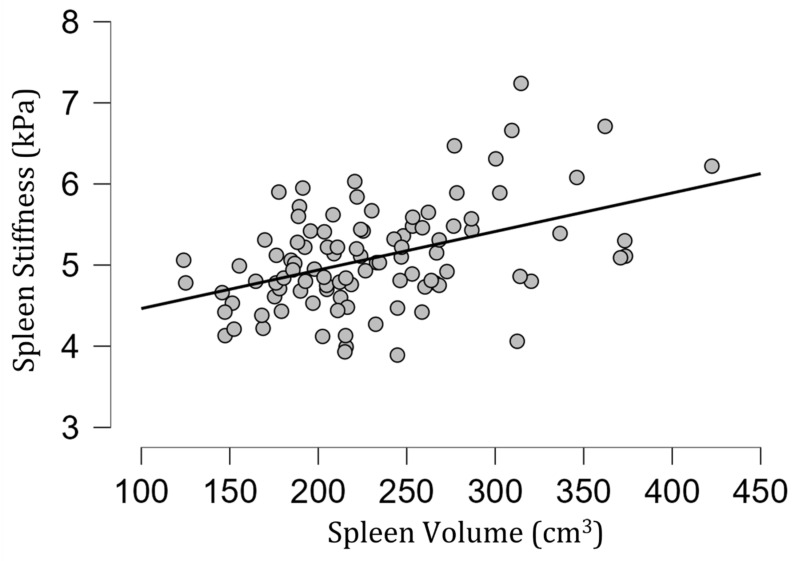
A scatter plot of spleen stiffness (in kilopascals) versus spleen volume. A statistically significant positive correlation between these two parameters was found (rho = 0.40, *p* < 0.001).

**Table 1 diagnostics-13-02738-t001:** A brief summary of the key results of this work.

Descriptives			
Parameter		Range	Mean [SD]
Spleen Stiffness	Male	3.99–7.24 kPa	5.28 [0.78] kPa
	Female	3.89–6.03 kPa	4.98 [0.51] kPa
	Whole Group	3.89–7.24 kPa	5.09 [0.65] kPa
BMI		16.85–25.76 kg/m^2^	21.33 [2.16] kg/m^2^
Spleen Volume		123.94–422.43 cm^3^	231.13 [58.44] cm^3^
Spleen Fat Content		1.15–3.27%	2.04 [0.39]%
**Correlation Results**			
Spleen Stiffness and Sex		*p* = 0.12	
Spleen Stiffness and Spleen Fat Content		rho = −0.13; *p* = 0.20	
Spleen Stiffness and R2*		rho = 0.03; *p* = 0.74	
Spleen Stiffness and BMI		rho = 0.09; *p* = 0.36	
Spleen Stiffness and Liver Stiffness		rho = 0.01; *p* = 0.95	
Spleen Stiffness and Spleen Volume		rho = 0.40, *p* < 0.001	

## Data Availability

The datasets generated or analyzed during the study are available from the corresponding author upon reasonable request.

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
