# Peer review of "Evaluation of Spleen Stiffness in Young Healthy Volunteers Using Magnetic Resonance Elastography"

_diagnostics, 2023, doi:10.3390/diagnostics13172738_

Round 1
Reviewer 1 Report
This is an interesting, well-written study aiming to present normative data for MRE-assessed stiffness measurements of the spleen in young adults. The methods, results, and discussion are nicely presented and I feel that it is an important work to the field considering the current limitations in the literature.
My only suggestion is to reduce a little the Introduction, as it is too long and addressing with too many details the technical aspects of MRE, whose review is not the main objective of this work. I also suggest including a reference/ citation for the following statement in Introduction: "MRE was developed by 43 researchers at the Mayo Clinic in Rochester, USA, during the 1990s."
I have no other remarks.
Reviewer 2 Report
General Comments
The presented study has some advantages, such as identified the normative data for MRE-based spleen stiffness in a group of healthy young subjects and established the correlation between spleen MRE stiffness and other parameters of the clinical history and examination. The merit of the work is that the results were well structured, and the corresponding figures were clear and concise. However, I have some suggestions which would significantly improve this manuscript.
Major Comments
- The Introduction section is a description of the characteristics and advantages of MRI elastography. But the importance of spleen studying is not described at all. This needs to be fixed.
- Line 114 ‘However, there is a need to establish reference data for these studies since information on MRE-based splenic stiffness in healthy individuals is currently limited.’ and Line 251 ‘The advantage of the current research is the largest study group compared with all other reports on this topic, available in the literature.’ However, in the Discussion section, Line 216 ‘Another study reported a mean splenic stiffness value of 5.2 ± 1.5 kPa in a sub-group of 115 patients not displaying symptoms of chronic liver disease.42 This value is comparable to our own findings but was obtained based on a retrospective analysis of medical records.’ If you cite a retrospective study with more than hundred cases, in addition to other studies, then the statement of ‘information is limited’ is debatable. Also, the sample size in study [42] is larger than yours. What are the benefits and novelty of your study?
- Line 207 ‘These results show that the normal spleen is stiffer than the normal liver, pancreas, prostate, and uterus.‘ and Line 242 ‘Hence, there is a limited possibility to compare or extrapolate the findings obtained by using different diagnostic modalities.’ It is necessary to speak only about the results of MRI elastography. For other methods of calculating stiffness at the macro-, micro- and nano-scale resolution, these regularities may not be relevant. For example, you can view the results of optical methods [Kennedy B. et al. The emergence of optical elastography in biomedicine. Nature Photonics 2017. 10.1038/nphoton.2017.6 ; and Zaitsev V.Y. et al. Strain and elasticity imaging in compression optical coherence elastography: The two-decade perspective and recent advances. Journal of Biophotonics 2021. 10.1002/jbio.202000257] or atomic-force microscopy [ZemÅ‚a J. et al. Atomic force microscopy as a tool for assessing the cellular elasticity and adhesiveness to identify cancer cells and tissues. Seminars in Cell & Developmental Biology 2018. 10.1016/j.semcdb.2017.06.029 ; and Guimarães, C.F. et al. The stiffness of living tissues and its implications for tissue engineering. Nature Reviews Materials 2020. 10.1038/s41578-019-0169-1] by stiffness study of biological tissues.
Minor Comments
- Line 99 ‘It has the potential to serve as an alternative to expensive and invasive liver biopsy.’ This controversial statement, because MRI does not provide information about the tissue morphology, which is obtained by histological examination.
- Line 143 ‘Only volunteers with liver stiffness <2.9 kPa…’ I did not see any explanation for this choice.
Round 2
Reviewer 2 Report
This manuscript has been improved to publication in Diagnostics.